# Total Triterpenes, Polyphenols, Flavonoids, and Antioxidant Activity of Bioactive Phytochemicals of *Centella asiatica* by Different Extraction Techniques

**DOI:** 10.3390/foods12213972

**Published:** 2023-10-30

**Authors:** Rasangani Sabaragamuwa, Conrad O. Perera

**Affiliations:** 1Food Science Programme, School of Chemical Sciences, University of Auckland, Private Bag 92019, Auckland 1142, New Zealand; rasangi.sabaragamuwa@gmail.com; 2Department of Food Science and Microbiology, School of Science, Auckland University of Technology, Auckland 1142, New Zealand

**Keywords:** *Centella asiatica*, extraction technique, green technology, microwave, phytochemicals, subcritical water, ultrasound

## Abstract

Obtaining phytochemical-rich plant extracts from natural products where the active ingredients are present in comparatively low levels in the tissue matrix is the critical initial step of any chemical analysis or bioactivity testing. The plant *C. asiatica* is rich in various phytochemicals, the major constituents being triterpenes and flavonoids, as well as other polyphenols, leading to a number of bioactivities. In this study, an attempt was made to achieve several green technology principles, while optimizing the extraction method for the efficient extraction of bioactive compounds from *C. asiatica*. Soxhlet extraction (SE), ultrasound-assisted extraction (UAE) with low-frequency sonication, microwave-assisted extraction (MAE) using a closed-vessel microwave digestion system, and subcritical water extraction (SWE) in a high-pressure reactor were employed to extract the bioactive compounds. The solvent system, extraction time, and solid-to-solvent ratio were varied to optimize the extraction. UAE gave the best extraction yield, while MAE gave similar results, with a solid-to-liquid ratio of 1:25, a binary solvent system of 9:1 methanol to water (*v*/*v*), and a 20 min extraction time for the extraction of triterpenes, including madecassoside, asiaticoside, madicassic acid, and asiatic acid. Investigation of different solvent systems based on water and methanol also revealed information on the extraction behavior of total triterpene content (TTC), total polyphenolic content (TPC), total flavonoid content (TFC), and the variations in the antioxidant capacity of the extracts. In this study, it was evident that UAE and MAE offer more efficient and effective extraction of bioactive compounds in terms of extraction yield, time, and minimal solvent and energy use. Furthermore, the results showed that the different solvent ratios in the extraction mixture will affect the extraction of bioactive compounds, and a binary solvent system with a combination of methanol and water was the most efficient for the studied compounds in *Centella asiatica*.

## 1. Introduction

Plant extracts are rich in an array of phytochemicals with diverse properties. Natural bioactive compounds show a broad diversity of structures and functionalities due to the wide range of functions they perform in plants, especially in response to the interaction between the plant and its environment. Plant bioactive compounds are produced as secondary metabolites that have functions in survival and subsistence but not in plant growth. These elicit pharmacological or toxicological effects in humans and animals [1]. According to Croteau, Kutchan, and Lewis (2000) [2], these are divided into three main categories: (a) terpenes and terpenoids, (b) alkaloids, and (c) phenolic compounds. The concentration of these compounds present in the plant tissues may vary, usually within very low levels. These bioactive compounds are present in different parts in the plant matrix together with other compounds such as conjugates or mixtures.

Extraction is the critical initial step of any chemical analysis or bioactivity testing, yet this step has in some cases become the bottleneck of the whole process [3]. Selection of the best extraction solvent, the extraction technique, and control of other parameters, such as time, temperature, particle size of the raw materials, and solid-to-liquid ratio, are important in extracting specific bioactive compounds.

Depending on the varying nature of the natural source, selection of the specific extraction technique from among the different techniques available should be performed appropriately. However, the objectives of these techniques are more or less the same: (1) to extract targeted bioactive compounds from a complex plant sample, (2) to increase the selectivity of analytical methods, (3) to increase the sensitivity of the bioassay by increasing the concentration of targeted compounds, (4) to convert the bioactive compounds into a more suitable form for detection and separation, and (5) to provide a strong and reproducible method that is independent of variations in the sample matrix [4]. The more conventional technique, as in solid-liquid extraction, is to bring the plant materials in contact with a solvent. For hundreds of years, conventional or classical techniques such as maceration, hydro-distillation, and Soxhlet extraction have been used to extract bioactive compounds from plant tissues [5]. In contrast, novel extraction techniques that are considered effective and more environmentally friendly are widely used at present, at both the laboratory and industrial scale, in nutraceuticals, pharmaceuticals, food additives, and many other sectors. These offer the benefits of lower extraction time, lower use of solvents, use of lower temperatures, higher selectivity, and less energy, while giving a higher comparative extraction yield [5,6]. These non-conventional methods enhance the performance with an assisted technology, such as ultrasonic waves, pressure, or microwaves. Altering of these physical parameters of the treatment mainly aims at cell wall rupture or deterioration, which enhance the mass transfer and effective mixing due to the exposure of cytoplasmic contents to the solvent [6,7].

Three non-conventional extraction techniques were used in this study. In ultrasound-assisted extraction (UAE), ultrasonic waves are used to cause cavitation, which leads to the implosion of bubbles in the medium. Cavitation induces collisions and shear in the reaction mixture, which leads to the disruption of solid particles, increasing the mass transfer rate and the penetration of solvents into biomass [8]. The microwave-assisted extraction (MAE) technique assists the extraction using microwaves. The energy associated with microwave irradiation is converted to thermal energy due to intracellular moisture evaporation, resulting a pressure build-up and rupture of cells, thus releasing bioactive compounds [9]. Subcritical water extraction (SWE) uses subcritical water under high temperature and high pressure, which change the polarity and dielectric constant of water, thus contributing to a better extraction process, improving the mass transfer efficiency of the extracts [10].

The plant *Centella asiatica* is rich in various phytochemicals, the major constituents being triterpenes, flavonoids, and essential oils [11]. These phytochemicals are purported to possess neuroprotective effects. The definition of a neuroprotectant includes the following key elements: prevention of neuronal death by inhibiting one or more of the pathophysiological steps in the process following damage to the nervous system, protection against neurodegeneration and neurotoxins, and interventions to slow down or halt the progression of neural degeneration [12]. Research evidence emphasizes the multifaceted nature of the *C. asiatica* herb as a potential neuroprotective agent in accordance with the above definition. Cooper and Ma (2017) [13] have also indicated *C. asiatica* as a potential phytopharmaceutical as it exhibits comprehensive neuroprotection via various effects such as reducing oxidative stress, inhibiting enzymes, and preventing the formation of amyloid plaques in AD patients.

Pentacyclic triterpenoid saponins, which are collectively known as centalloids, are considered one group of secondary metabolites and are mainly responsible for the biological activities ascribed to the pharmacological activities of *C. asiatica* [14,15,16]. Jäger, et al. (2009) [17] mentioned that besides other phytochemicals, pentacyclic triterpenes from the lupane, oleanane, and ursane groups have been identified as active substances from plant sources of the healthy Mediterranean diet. The triterpene group of compounds includes both sterols and triterpenes and are accumulated as glycosides (saponins) in extensive amounts in some plants. Saponins are classified according to their aglycone skeleton. A pentacyclic triterpene structure consists of a C30 skeleton. *C. asiatica* triterpenes can be subdivided into two groups according to methyl substitution patterns on the C19 and C20 as oleanane and ursane [16]. The most prominent triterpenes in *C. asiatica* are madecassoside, asiaticoside, and their sapogenin triterpene acids (madecassic and asiatic acid). Thus, they are considered signatory biomarkers of the metabolome of this plant.

This paper describes the investigation of different extraction methods for the efficient extraction of potential neuroprotective bioactive compounds from *C. asiatica*. Four extraction techniques were implemented, namely the conventional Soxhlet extraction (SE) and three relatively novel methods, ultrasound-assisted extraction (UAE), microwave-assisted extraction (MAE), and subcritical water extraction (SWE), which are considered as green extraction technologies. In addition, the solvent system, extraction time, and solid-to-solvent ratio were varied to optimize the extraction. The green extraction concept is an innovative current topic in natural product research [18,19]. In this study, an attempt was made to achieve several green technology principles, as stated by Chemat et al. (2019) [18], while optimizing the extraction method. Reversed-phase high-performance liquid chromatography coupled with diode array detection (RP-HPLC-DAD) was used to compare the presence and the total amount of the four major triterpenes considered as signatory biomarkers of *C. asiatica* extracts.

The aim of this study was to analyze the total triterpenes, total phenolic content, flavonoid content, and free radical scavenging activity of bioactive phytochemicals of *Centella asiatica* under selected extraction conditions and comparing the extraction behavior of different solvent systems.

## 2. Materials and Methods

### 2.1. Plant Materials

*C. asiatica* plant samples were obtained from a cultivated plot of a home garden in Auckland, New Zealand. A voucher specimen of the samples was deposited in Auckland War Memorial Museum (AK 370706). The species was authenticated by verification of morphological characteristics and by DNA molecular analysis. The morphological characteristics were verified by experts, comparing their observations against a reference collection housed at Auckland War Memorial Museum herbarium, as well as comparing against Flora of New Zealand (*Centella uniflora* [20]) and Flora of China (*Centella asiatica* [21]) accounts. A PCR-amplified internal transcribed spacer (ITS) region was sequenced and compared with known sequences in GenBank to identify and confirm the species of the specimen [22].

### 2.2. Chemicals

Methanol (HPLC grade), acetonitrile (HPLC grade), ethanol (HPLC grade), and formic acid (99%) were purchased from Sigma Aldrich, Auckland, New Zealand. Triterpene standards (asiaticoside, madecassoside, asiatic acid, madicassic acid) were obtained from AK Scientific, Union City, CA, USA. Pure water was used from Millipore 18.2 water purification system (Millipore Corporation, Burlington, MA, USA). All the chemicals for total phenolic, total flavonoid, and DPPH radical scavenging assay, including Folin–Ciocalteu phenol reagent, 1,1-diphenyl-2-picrylhydrazyl (DPPH), aluminum chloride, catechin, and gallic acid, were purchased from Sigma Aldrich, Auckland, New Zealand. Sodium carbonate, sodium hydroxide, and sodium nitrite were from Scharlau, Barcelona, Spain.

### 2.3. Sample Preparation

Freshly harvested plant samples were cut at 1 cm below the petiole and air dried in single layers using an electric dehydrator (Ezdri Ultra FD100, Hydraflow Industries Ltd., Tauranga, New Zealand), with a continuous air flow at 40 °C for 18 h. The dehydrator consisted of four drying racks, and the weight reduction due to moisture removal in each rack was recorded to ensure uniform drying conditions. Air drying at 40 °C is also recommended in European Pharmacopeia for *C. asiatica* leaves as a pre-sample preparation method before extraction and analysis [23]. Dried samples were immediately ground to a fine powder using a coffee grinder (Breville Inc. Auckland, New Zealand). Powdered samples were packed in nitrogen-flushed, screw-thread amber glass vials (4-dram capacity) and stored under −20 °C until used in further experiments.

### 2.4. Moisture Content Determination

The moisture content of the fresh leaves was determined by the vacuum oven drying method according to AOAC method 934.0 (AOAC, 1983). The dry matter content of the samples was calculated based on the moisture content. This was used in the calculation of the yield of constituents on a dry matter basis.

### 2.5. Color of the Samples

The color of the samples was also determined for the fresh samples and dry powder using CIE (Commission Internationale de L’Eclairage) L*a*b* color space measured by colorimeter (CR-300, Minolta, Konica) with diffuse illuminator, 8 mm aperture, and 0 degrees observer. Measurements were taken in triplicate.

### 2.6. Selection of Optimum Extraction Conditions

The extraction was optimized based on the yield of four major triterpenes of *C. asiatica*, as the response variable, quantified by HPLC. The experiments were designed as single-factor experiments, changing one independent variable at a time, with four variables: extraction technique, solvent system, solid-to-liquid ratio, and extraction time. The levels of each factor were chosen based on previous studies and preliminary experiments. All extractions were done in triplicate.

Four extraction techniques, microwave-assisted extraction (MAE), ultrasound-assisted extraction (UAE), Soxhlet extraction (SE), and subcritical water extraction (SWE), were performed.

Methanol was selected as the solvent of choice for initial extraction experiments, considering the polarity of the major compounds [24], polarity index, and dielectric constant of the solvent, as well as the previously published literature [25,26]. A sample of 2 g of dried *C. asiatica* was extracted in 50 mL of methanol, except when the solid:liquid ratio was being investigated.

After selecting the extraction techniques yielding comparatively higher triterpene content in the extract, six solvent systems were investigated, based on methanol, ethanol, water, and the different ratios of mixtures of the same. To reduce the number of treatment levels of this factor, the polarity and the dielectric constant of the solvents as well as the previously published literature were taken into consideration as mentioned earlier. In the last step, the two levels of solid:liquid ratios and three levels of extraction time periods were examined. The factors investigated and their levels are given in Figure 1.

#### 2.6.1. Soxhlet Extraction

Soxhlet extraction was carried out on the dried powder using a Soxhlet apparatus (Soxtherm, Gerhardt UK Ltd., Backley, UK). The extraction temperature was approximately 64.7 °C, since this is the boiling point of methanol. An extraction duration of 8 h was used as this is reported as the optimal condition for Soxhlet extraction for *C. asiatica* [23,27]. The collected extract at the end was evaporated to dryness by a vacuum rotary evaporator, and the residue was dissolved in 10 mL of methanol. This extract was filtered using 0.22 μm PTFE filters (MicroAnalytix, Auckland, New Zealand) and proceeded to HPLC analysis.

#### 2.6.2. Microwave-Assisted Extraction (MAE)

Microwave-assisted extraction was carried out using a microwave digestion system (Ethos, SK-15 rotor, Milestone, Italy), consisting of closed vessels (PTFE-TFM) with safety shields on a rotating carousel. Even though the microwave power was set at 1800 W at the beginning, only 6–8 vessels in an alternative arrangement in the carousal were utilized for the samples each time; thus, the microwave power only reached up to 600 W under real operational conditions.

The microwave power, the internal pressure, and the inside and outside temperature of the vessels were shown on the display. The temperature sensor monitored the real-time temperature change inside the vessels.

The extraction procedure was programmed using the touchscreen terminal (Table 1). Methanol was used as the extraction solvent at a 1:25 (g:mL) solid-to-solvent ratio. The powder samples were suspended in the respective solvent and transferred into the extraction vessels. After the end of the given holding time at the set temperature, the vessels were allowed to cool down before opening. The extracted samples were centrifuged (Heraeus vacutherm, Themo Scientific, Auckland, New Zealand) at room temperature at 2465.19× *g* for 10 min, and the supernatant was collected and evaporated under a gentle nitrogen flow. The residue was dissolved in 10 mL of methanol and filtered using 0.22 µm PTFE filters before analysis by HPLC.

#### 2.6.3. Ultrasound-Assisted Extraction (UAE)

A Sonic Rupture 250 (Omni International, Kennesaw, GA, USA) instrument was used for ultrasound-assisted extraction. This instrument was attached to a sonication probe (exponential taper horn), operating at the frequency of 20 kHz, which is considered low-frequency sonication. The energy output was set at 50% amplitude. The energy input of the instrument was 250 W theoretically.

Dried *C. asiatica* powder (2 g) was suspended in methanol (50 mL) and mixed well, and the ultrasonic probe with 1.9 cm diameter was positioned 1 cm below the solution. The sonication probe was a solid tip probe since low-surface-tension solvents were used in the extraction. A narrow-mouth vessel (100 mL, wide-neck Erlenmeyer flask with 50 mm neck diameter) was used to ensure that the whole sample was mixed and treated uniformly. Sonication was performed with a 90% pulse rate in an ice bath to prevent temperature increase during the extraction and to increase the intensity of ultrasonic exposure. Initially, the sample was sonicated for 30 min with pulsating (effective sonication time was around 20 min). The sonicated samples were centrifuged for 10 min at room temperature at 2465.19× *g*, and the supernatant was collected and filtered with 0.22 µm PTFE filters before analysis by HPLC.

#### 2.6.4. Subcritical Water Extraction (SWE)

Subcritical water extraction of triterpenes from air-dried *C. asiatica* leaf powder was carried out in a 1 L Parr Reactor (Series 4540 high-pressure reactor; Parr Instrument Company, Moline, IL, USA). Instead of using organic solvents, high-temperature water below supercritical pressure was used as the extraction media in SWE. The dried leaf powder was added at a 5% concentration to the reactor following the standard operating procedure given. Since the capacity of the reactor was 1 L, 30 g of powder and 600 mL of water were added to reach a 5:100 g/mL (solid/liquid) ratio. The pressure vessel was initially purged with nitrogen gas to remove the oxygen present. Additional nitrogen gas was applied to pressurize the vessel, and the initial pressure applied was 5 MPa (50 bars). The operational pressure was held at 7.5 ± 0.3 MPa (75 ± 3 bars). Two extraction temperatures were investigated, one at 175 °C and the other at 200 °C. The extraction times were varied from 10 to 90 min. During the continuous operation, 15 mL of extracts were taken at 10 min intervals and vacuum filtered using filter paper (Whatman No.1), and filtrates were stored at refrigeration conditions in UV-safe tubes, which were flushed with nitrogen gas until further analyses.

### 2.7. High-Performance Liquid Chromatography (HPLC) Analysis

The HPLC analyses of four major triterpenes of the extracts were performed by adopting two reported methods in literature [27] with some variation in the column dimensions and HPLC system used. Four major triterpenes, asiatic acid, asiaticoside, madecassic acid, and madecassoside, were identified and quantified against external standards with reversed-phase (RP) HPLC. An Agilent 1200 HPLC system with a diode array detector G1315A (detection at 205 nm), vacuum degasser G1322A, quaternary pump G1311A, autosampler G1313A, and column oven G1316A were used for the analysis. Agilent ChemStation (B.04.03) software was used for data acquisition and analysis. A Synergi 4 µm, 80 Å, Fusion-RP C18 analytical column with dimensions of 150 × 4.6 mm was used for the chromatographic separation with gradient elution. Pure water (A) and acetonitrile (B) were used as mobile phases starting with 80% A and 20% B, then changing it consequently to 65% A at 15 min, 35% A at 30 min, 65% A at 40 min, and coming back to 80% A and equilibrating for 5 min, with a flow rate of 1 mL/min. The sample injection volume was 20 µL. The column temperature was maintained at 30 °C. Calibration curves were established with five concentrations of each standard in the 0.1–4 mg/mL range.

Standard stock solutions were prepared by dissolving 10 mg of each triterpene in 2 mL of methanol. Standard stock solution mixtures were also prepared and stored under −20 °C. These stock solutions were diluted appropriately to obtain calibration standards as needed.

### 2.8. Particle Size Measurement

The particle size distribution of the initial raw material before extraction (dry powder) was analyzed by laser diffraction technique using a Malvern Mastersizer 2000 instrument (Malvern Instruments Limited, Malvern, UK). The estimates of particle size distribution and diameter were later compared with the same estimates of the residues after ultrasound and microwave extractions.

### 2.9. Total Phenolic Content (TPC), Total Flavonoid Content (TFC), and DPPH Radical Scavenging Activity (RSA_DPPH_)

The reducing capacity and total phenolic content of the extracts were estimated using Folin–Ciocalteu (F-C) method originally reported by Singleton and Rossi (1965) [28] with some modifications [29].

An aliquot of 100 µL of each sample supernatant, standard, or methanol blank was added to duplicate 2 mL microtubes. Then, 200 µL of 10% (*v*/*v*) F–C reagent was added and vortexed thoroughly. The F–C reagent should be added before the alkali to avoid the air oxidation of phenols. Next, 800 µL of 700 mM Na_2_CO_3_ was added into each tube and incubated at room temperature for 2 h. An aliquot of 200 µL each of the sample, standard or blank, from the assay tubes was transferred to a clear 96-well microplate, and the absorbance was read of each well at 765 nm. Gallic acid (0–1000 μg/mL) was used for the calibration curve. The results were expressed as Gallic acid equivalent (mg GAE/100 g DW) and calculated as mean value ± SD (*n* = 6).

Total flavonoid content was estimated by the aluminum chloride method [30]. In a 96-well plate, 100 µL of distilled water, 10 µL of 5% NaNO_2_, and 25 µL of sample/standard/blank were added. Then, 15 µL of 10% AlCl3 was added after 5 min. After 6 min, 50 µL of 1 M NaOH and 50 µL of water were added. The plate was shaken for 30 s, and the absorbance was measured at 510 nm for total flavonoid content. A standard curve was created from the blank-corrected absorbance at 510 nm of catechin standards (0–1000 μg/mL). The total flavonoids were calculated as catechin equivalents (mg of CAE/100 g of dry weight) using the regression equation.

DPPH radical scavenging activity (RSA_DPPH_) was measured using the method of Brand-Williams et al. (1995) [31] with modifications. A methanolic solution of DPPH (0.05 mg/mL) was made freshly before each experiment. Aliquots of 10 μL of samples were pipetted into the 96-well plate. Then, 200 μL of DPPH solution was added to each well, and the absorbance was recorded at 517 nm against methanol and water as the blank (negative control) after incubation in the dark at 20 °C for 60 min. The percentage of DPPH radicals scavenging activity was calculated in terms of Trolox equivalent based on a calibration curve with Trolox concentration ranging from 1 mM to 25 μM.

### 2.10. Statistical Analysis

The variance in triterpene yield under different techniques was analyzed by one-way ANOVA followed by Tukey’s test to compare means, using GraphPad Prism version 8.3.1 for macOS (GraphPad Software, La Jolla California USA, www.graphpad.com, accessed on 15 July 2023). The same statistical tool was used to investigate the variation in total triterpenes, total polyphenols, total flavonoids, and antioxidant capacity of extracts under different binary solvent systems. The statistical analysis was performed by IBM SPSS software (version 26).

## 3. Results

### 3.1. Moisture Content Determination, Color Measurements, and Pre-Sample Preparation

Based on triplicate samples analyzed by vacuum drying method, the average moisture content of the fresh *C. asiatica* leaves was calculated as 84.30 ± 1.70%, so the percentage of dry matter of the leaves was calculated as 15.70%. This value was taken in further calculations of triterpene yields in the extracts. The color measurements showed that the air drying method used has not affected the color of the leaves since there was only a slight change in the L*a*b* values (no significant difference according to the paired *t*-test at 0.05 alpha level). This can be taken as an indication of fewer changes to the original plant material after drying. Color measurements of the samples before and after drying are provided in Appendix A.

Air drying of fresh leaves prior to the extraction is important in various aspects. Drying will prevent tissue deterioration and alterations to the phytochemical composition (except for the essential oils) due to the enzymatic action and microbial activity [32]. Drying also removes excess water, making the constituents more concentrated. This will also lead to reproducibility and accuracy of the results, in the experiment and in the calculations, generating fewer errors since the mass was taken on a dry matter (DM) basis. The effect of moisture content of the raw materials in extraction is emphasized in the literature [5,33]. Therefore, controlling this variable by preparing samples with the same standard procedure in pre-sample preparation will contribute to reduced deviations due to the differences in the moisture content. Furthermore, grinding of the samples to a fine powder enables more efficient extraction. Durling et al. (2007) [34] have studied the effect of the particle size of the raw material on the extraction efficiency. They reported that the particle size controls the mass transfer kinetics and the access of the solvent to the soluble components in the extraction of bioactive compounds. Thus, higher extraction efficiencies are achieved with smaller particle sizes, due to the increase in mass transfer surface area [35]. Comparison of the particle size distribution of initial raw material (dry powder) and the residues after the extraction methods, UAE and MAE, are shown in Appendix A.

The average diameter of the particle size (D [2,3] in µm) of the dried and grounded *C. asiatica* powder was 45.70 ± 0.91 µm as measured by laser light scattering. This was reduced to 42.66 ± 0.04 µm after MAE extraction and after UAE extraction, it was reduced to 35.21 ± 0.33 µm.

Reduction of particle size indicates the effective rupturing of cells to release more bioactive compounds. Moreover, the smaller the particle size, the larger the surface area of the total mass enabling better mass transfer during extraction. Figure 2 shows the statistical comparison of the extraction yield of triterpenes. Although there is no significant difference between the MAE and UAE yield, UAE has provided a slightly higher yield of triterpenes. The smallest particle diameter of the residue after UAE shows that it is most effective in rupturing the cell membranes to extract the bioactive compounds.

### 3.2. Selection of Optimum Extraction Conditions for Triterpenes

The total triterpene yield showed a significant variation according to the extraction technique used (Figure 2).

The HPLC chromatogram of the Soxhlet extract visibly showed only two triterpene peaks: asiaticoside and madicassoside. The energy use was higher, since the extraction time was longer, and moreover, the amount of solvent used was relatively higher in SE than in UAE or MAE. SWE is a solvent-free technique that has been reported in successful bioactive compound extraction in previous studies [36,37,38]. However, in this study, it was observed that none of the major triterpenes were present in the subcritical water extract at any of the extraction conditions investigated. Kim et al. (2009) [39] have reported the use of subcritical water in the extraction of two triterpenes: asiaticoside and asiatic acid from *C. asiatica*. In that study, they used a slightly different custom-made setup, but with the same operational principle. They used an extraction time of 5 h at a temperature of 250 °C under 40 MPa pressure as optimum conditions. Their study has shown that SWE under the conditions used could extract a higher amount of the two triterpenes than by extracting with methanol at room temperature. However, in the same study, methanol at its boiling point temperature has provided the highest yield [39].

The dielectric constant (ε) of subcritical water approaches 30 at 250 °C temperature, which is closer to that of methanol (ε = 33 at 25 °C; 27 at 65 °C). Theoretically, at subcritical temperatures, water will have reduced polarity, facilitating the dissolution of less polar compounds and enhancing their extraction efficiencies [40]. Kim et al. (2009) [39] reported an interesting finding, which we also suspected in our study: that the triterpenes were not present in the filtrate but in the filter cake. We used the Whatman No. 1 filter (pore size is 11 µm) and analyzed the filtrate to determine the absence of any triterpenes. Since the water was depressurized and cooled to room temperature after the subcritical water extraction, the change in the polarity of water back to its initial condition may have reduced the solubility of triterpenes in water. Kim et al. (2009) [39] have also indicated a possible ‘precipitation’ of these triterpenes. Since filtering is essential after the SWE prior to the injection into the HPLC system for analysis, they have suggested dissolving the filter cake in methanol again to extract the triterpenes present there. In our study, we did not do an in-depth investigation into this, considering the lengthy procedure and other practical difficulties of the extraction process. Therefore, only the comparison of the rest of the three extraction techniques is discussed further in this study.

Among the investigated extraction techniques, ultrasound- and microwave-assisted methods resulted in the highest yield of total triterpenes. UAE showed slightly higher values of triterpene yield, yet no statistically significant difference was present between MAE and UAE values (Figure 2 and Table 2).

Both UAE and MAE gave similar HPLC chromatograms, as shown in Appendix A, indicating that they both extracted the same triterpenes at similar levels. The analysis of total triterpene yields with two different solid:liquid ratios revealed that a 1:25 solid-to-liquid ratio resulted in the best extraction compared to the 1:10 or 1:50 ratios. Methanol was superior to ethanol as an extraction solvent in terms of the extracted triterpene yield. The effect of varying methanol concentrations was also investigated, and the results are discussed in the next section. There was no significant increase in the extracted triterpene yield with an increase in the extraction time from 20 to 30 or 45 min. All these parameters were statistically compared for significance at a 95% level of confidence.

These results were comparable with two similar studies [41,42]. In the current study, the focus was to optimize the extraction parameters in terms of higher triterpene yield, less energy consumption, and reduced solvent use. Consequently, ultrasound-assisted extraction for 20 min with methanol as the extraction solvent and a 1:25 solid-to-liquid ratio were selected as the best extraction method.

### 3.3. Investigation of Total Triterpene Content (TTC), Total Phenolics Content (TPC), Total Flavonoids Content (TFC), and DPPH Radical Scavenging Activity (RSA_DPPH_) under Different Solvent Systems

The composition of bioactive compounds in plant extracts may vary depending on their chemical characteristics (such as degree of polarity) and the locality in the plant matrix. Binary solvent systems are reported to result in more efficient extraction of phytochemicals than mono-solvent systems [25,43]. On the other hand, different solvents have been used in bioactivity testing and phytochemical characterization of *C. asiatica* plant in different studies, resulting in significant differences in the amounts and composition of phytochemicals as well as their bioactivities compared to what would be present due to other variables such as geographical location or chemotype variations [16,42,44,45,46,47,48,49]. Methanol and ethanol have been the solvents of choice in most of these studies in mono or binary mixtures. In the current experiment, different ratios of methanol and water mixtures were used in the extraction starting from 60% (*v*/*v*) methanol. Only two glycosides, madicassoside and asiaticoside, were quantified as TTC for comparison purposes since these two glycosides were present in sufficient quantities, unlike their aglycone counterparts. The results indicated that the individual glycosides as well as the total glycoside content gradually increased with the increase of methanol percentage in the extraction solvent (Figure 3). However, 90% methanol extract yielded a slightly higher amount of total triterpenes (144.19 ± 0.04 mg/g DW) compared to the 100% methanol extract (140.91 ± 0.56 mg/g DW), which showed a statistically significant difference (*p* < 0.05). This result also agrees with the study of Shen et al. (2009) [42] investigating solvent mixtures of 30%, 50%, 90%, and 100% methanol.

It is reported in the literature that the dielectric constant of methanol-water mixtures is significantly affected by the concentration of methanol in the mixture [50,51]. The concentration at 90% methanol (*v*/*v*) may present the optimal conditions for centalloids of *C. asiatica*, especially in terms of polarity.

TPC also showed a gradual increase with the increase of methanol percentage in the extraction solvent (Figure 4). However, the TPC values of 90% methanol extract and 100% extract did not significantly statistically differ at the 0.05 alpha level, while those for 70% and 80% methanol showed a variation. The total phenolic content is an index of antioxidant capacity measured via the reduction of the Folin–Ciocalteu reagent by phenolic compounds, resulting in the formation of a blue complex that can be spectroscopically measured at approximately 760 nm. This method measures the reducing capacity, and certain other compounds in plant extracts, such as ascorbic acids, reducing sugars, and proteins, can also interfere with the estimation [52]. Phenolics are compounds that possess one or more aromatic rings with one or more hydroxyl groups and are generally categorized as phenolic acids, flavonoids, stilbenes, coumarins, and tannins [53]. Terpenoids are not generally classified as phenolics [54]; however, they also possess structural and functional similarities to polyphenols. Since triterpenes are the major compounds present in *C. asiatica* extracts, they may also have contributed to the TPC estimate obtained. There was a statistically significant (*p* < 0.05) strong correlation between the measured TTC and TPC in extracts of different solvent systems according to the Pearson correlation analysis conducted (Table 3).

Flavonoids are a major subclass of total phenolics in plant extracts and are reportedly one of the major classes of phytochemicals present in *C. asiatica* extracts. Total flavonoid content (TFC) also displayed a similar behavior to TTC, showing an overall gradual increase with the increase of methanol percentage in the extraction solvent, except for the 90% methanol level, as described below (Figure 4).

However, the correspondence between TTC and TFC was less strong than that of TTC and TPC (Table 3). TTC gave the highest value at the 90% methanol level and showed significant differences among all the solvent systems investigated, while TFC also gave the highest and most significant increase at 90% methanol extraction. However, TPC values with 90% methanol and 100% methanol were not statistically significant (*p* < 0.05). These results indicate that 90% methanol can be used to extract all the major bioactive compounds, triterpenes, polyphenols, and flavonoids, from the dried leaves of *C. asiatica* at their optimum level.

DPPH radical scavenging activity (RSA_DPPH_) estimates, which can be used to examine the antioxidant capacity of extracts, are shown in Figure 4. All the bioactive compounds that have the potential to scavenge free radicals will reduce DPPH by donating a hydrogen atom. In this assay, RSA_DPPH_ estimates of the 80%, 90%, and 100% methanol solvent systems did not show a statistically significant difference among each other (at 0.05 alpha level), while lower solvent concentrations gave lower values.

Since it was anticipated that this free radical scavenging activity would be provided mainly by triterpenes, phenolics, and flavonoids, the correlation of TTC, TPC, and TFC with RSADPPH was explored. Pearson correlation analysis was performed to determine the correlation among these four parameters: TTC, TPC, TFC, and RSA_DPPH_ (Table 3). The correlation matrix showed that there are strong positive correlations between each pair of variables, at a 95% level of confidence. It is interesting to observe that there is a very strong positive correlation between DPPH radical scavenging activity and the other three estimates of TTC, TPC, and TFC, which implies that all of these bioactive compounds (triterpenes, phenolics, and flavonoids) contribute to reducing DPPH free radicals, indicating their contribution to antioxidant potential.

## 4. Conclusions

This paper investigated the potential application of recently developed green extraction techniques in the efficient extraction of bioactive compounds from the plant *Centella asiatica*. It was evident that UAE and MAE offer more efficient and effective extraction of triterpenes, the major bioactive compounds of the plant, in terms of extraction yield, time, and minimal solvent and energy use, while the Soxhlet and SWE resulted in poor to negligible extraction of the triterpenes. Investigation of different mono- and binary-solvent systems based on water and methanol revealed information on the extraction behavior of major bioactive compounds of this plant and the variation in the antioxidant capacity of the extracts. The extraction of bioactive compounds varied with different solvent ratios of the extraction mixture, mostly based on the polarity of the compounds. Methanol is a good organic solvent to extract most bioactive compounds; however, it is less polar compared to water. This study revealed that a binary solvent system using a combination of methanol and water led to a better extraction of the studied bioactive compound classes of *C. asiatica* in non-volatile or liquid fraction, including triterpenes, polyphenols, and flavonoids.

In addition to its main aim, the study also revealed interesting information on the effect of climatic conditions on compositional variation of bioactive compounds of the plant. It is well known that the secondary metabolites of plants respond to environmental conditions differently. *C. asiatica* samples harvested in September, in the springtime of New Zealand, yielded less triterpene glycoside content (73.54 ± 0.69 mg/g DW) compared to the samples harvested in April (140.91 ± 0.71 mg/g DW), in autumn. This emphasizes the importance of indicating the variability factors such as cultivar, geographic location, and harvesting season when comparing data from different studies.

## Figures and Tables

**Figure 1 foods-12-03972-f001:**
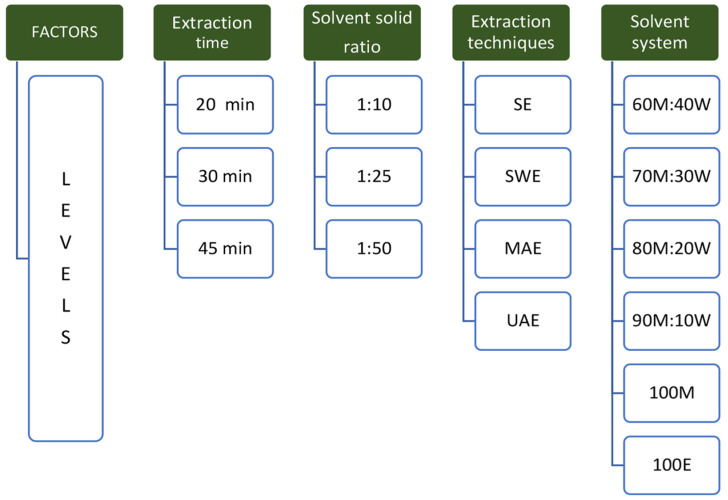
Factors and levels of studied variables. Key: SE—Soxhlet extraction, SWE—subcritical water extraction, MAE—microwave-assisted extraction, UAE—ultrasound-assisted extraction, M—methanol, W—water, E—ethanol.

**Figure 2 foods-12-03972-f002:**
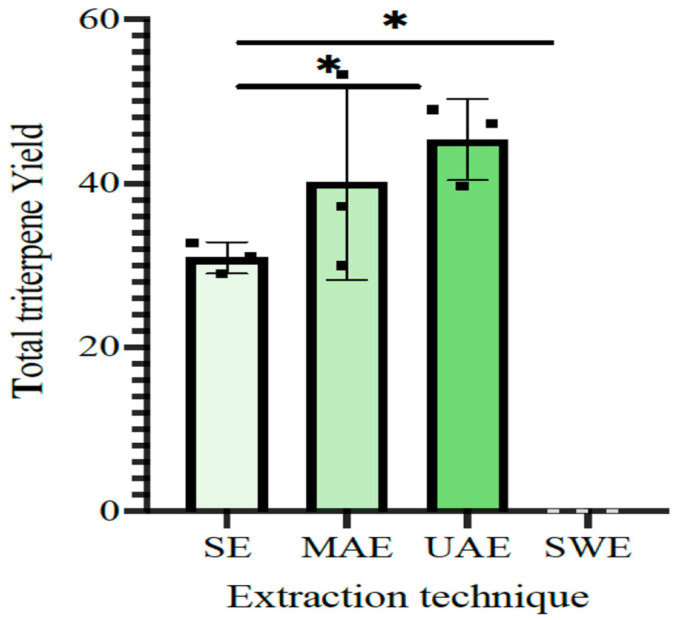
Comparison of total triterpene yield in three extraction techniques; SE, MAE, UAE, and SWE (* showing significant difference). SE—Soxhlet extraction, MAE—microwave-assisted extraction, UAE—ultrasound-assisted extraction, SWE—subcritical water extraction.

**Figure 3 foods-12-03972-f003:**
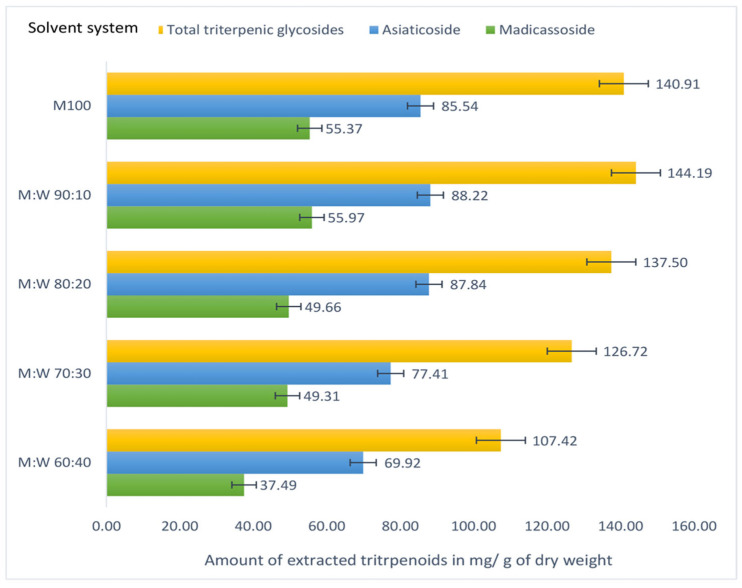
Change in triterpene glycoside contents with increasing methanol percentage in extraction solvent of microwave-assisted extracts of *C. asiatica* dry leaf powder.

**Figure 4 foods-12-03972-f004:**
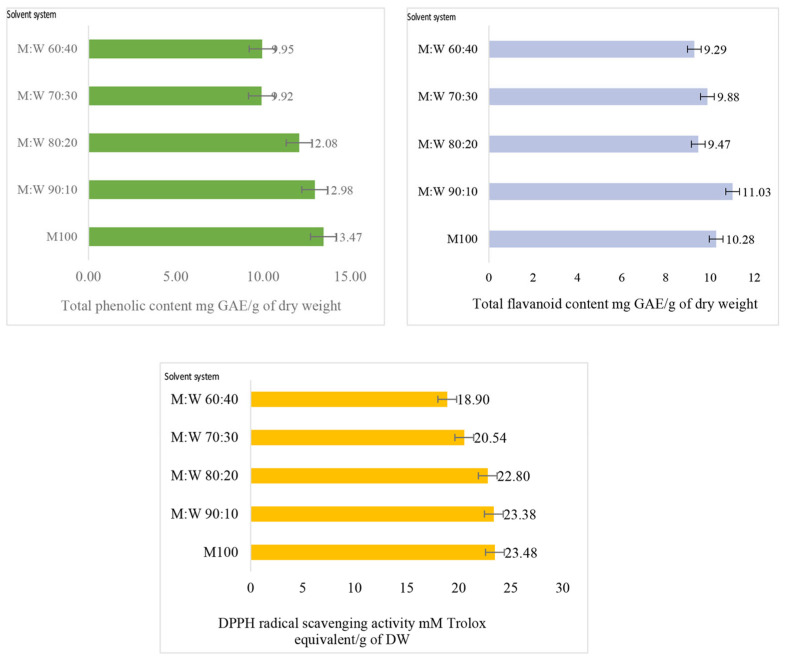
Change in total phenolic content, flavonoid content, and DPPH radical scavenging activity with increasing methanol percentage in extraction solvent of microwave-assisted extracts of *C. asiatica* dry leaf powder.

**Table 1 foods-12-03972-t001:** Microwave program.

Step		Time (min)	Power (W)	Temperature (°C)
01	Heating	05	0–600	20–70
02	Holding	20	0–180	70
03	Cooling	10	-	70–0

**Table 2 foods-12-03972-t002:** Comparison of the amounts of extracted triterpenes using different extraction methods.

Bioactive Compound	Extraction Yield (mg/g)		
	Soxhlet	Microwave-Assisted	Ultrasound-Assisted
Madicassoside	ND	25.05 ± 0.19 ^a^	23.95 ± 0.63 ^a^
Asiaticoside	19.93 ± 0.46 ^a^	48.49 ± 0.64 ^b^	51.58 ± 0.44 ^b^
Madicassic acid	11.68 ± 1.18 ^a^	5.91 ± 0.97 ^b^	6.82 ± 0.16 ^b^
Asiatic acid	ND	1.85 ± 0.47 ^a^	2.31 ± 0.47 ^a^
Total triterpenes	30.94 ± 1.92 ^a^	81.30 ± 1.08 ^b^	84.66 ± 1.32 ^b^

^a,b^ Different letters denote significant differences at 95% confidence interval in each parameter.

**Table 3 foods-12-03972-t003:** Pearson correlation table.

	TTC	TPC	TFC	RSA
TTC	1	0.86	0.83	0.98 **
TPC	0.86	1	0.98 **	0.94 *
TFC	0.83	0.98 **	1	0.89 *
RSA_DPPH_	0.98 **	0.94 *	0.89 *	1

* Significance at α = 0.05. ** Significance at α = 0.01.

## Data Availability

The data used to support the findings of this study can be made available by the corresponding author upon request.

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
