# Peer review of "Total Triterpenes, Polyphenols, Flavonoids, and Antioxidant Activity of Bioactive Phytochemicals of Centella asiatica by Different Extraction Techniques"

_foods, 2023, doi:10.3390/foods12213972_

Round 1
Reviewer 1 Report (Previous Reviewer 1)
Comments and Suggestions for Authors
Dear Authors, the manuscript "Total triterpenes, polyphenols, flavonoids, and antioxidant activity of bioactive phytochemicals of Centella asiatica by different extraction techniques " is quite interesting and worth investigation.
I agree with the authors answers to the reviewers and have no further comments.
Therefore, This paper was improved and perfect now.
Regards
Author Response
Thank you for your comments
Reviewer 2 Report (Previous Reviewer 2)
Comments and Suggestions for Authors
The article titled ‘Total triterpenes, polyphenols, flavonoids, and antioxidant activity
of bioactive phytochemicals of Centella asiatica by different extraction
techniques’ has been reviewed. The article shows the description of four different extraction techniques applied to Centella asiatica plant in order to obtain bioactive compounds. Major changes are necessary.
Introduction
The first time you name Centella asiatica in the text, the genus and specie are necessary. Please check in the introduction section.
‘The oldest technique’ change for ‘the more conventional technique’
The abbreviations of extraction techniques are named in the third paragraph, but in the last paragraph of the introduction section the full name is explained. Please indicate the abbreviation meaning the first time you write the words.
Next to references 18, 19 a ] is missing in the last paragraph
Materials and methods
Could you please reference the DNA extraction analysis? Or explain it.
Add the particle size after ground step.
Was the color samples carry out by duplicate? Please add
Did you check the color analysis only in the starting material? Why not in the extracts? Could be interesting since you are extracting some colorants.
Is the liquid chromatography method based on any paper? Please reference it
Check all the C. asiatica, they should be in italics, some of them are not
This kind of sentences ‘HPLC analyses of C. asiatica extracts are reported in the literature [26-32].’ Should be in the results and discuss section. Materials and methods section is only to describe the methods use. Please rephrase the sentence and indicate what methods you base your method. All the six methods (reference from 26-32 are necessary to reference?).
Results
Could you please include a picture in the Table ST1 to complement the results?
The figure S1 and S2 has not been attached to the submission or it is not available. Please include it.
Comments on the Quality of English Language
none
Author Response
The article titled ‘Total triterpenes, polyphenols, flavonoids, and antioxidant activity of bioactive phytochemicals of Centella asiatica by different extraction
techniques’ has been reviewed. The article shows the description of four different extraction techniques applied to Centella asiatica plant in order to obtain bioactive compounds. Major changes are necessary.
Introduction
The first time you name Centella asiatica in the text, the genus and specie are necessary. Please check in the introduction section.
Other than in the abstract, Centella asiatica is mentioned for the first time in the Introduction section; in the first paragraph of page 3. Full scientific name was given there (highlighted).
‘The oldest technique’ change for ‘the more conventional technique’
Changed
The abbreviations of extraction techniques are named in the third paragraph, but in the last paragraph of the introduction section the full name is explained. Please indicate the abbreviation meaning the first time you write the words.
Done
Next to references 18, 19 a ] is missing in the last paragraph
Corrected.
Materials and methods
Could you please reference the DNA extraction analysis? Or explain it.
Reference (22) included
Add the particle size after ground step.
It was included under the Results section (3.1- para 3) – highlighted
Was the color samples carry out by duplicate? Please add
Measurements were taken in triplicate. Added.
Did you check the color analysis only in the starting material? Why not in the extracts? Could be interesting since you are extracting some colorants.
We checked the color of the starting materials but not in the extracts as our aim was to to analyse, the total triterpenes, total phenolic content, flavonoid content, and free radical scavenging activity of bioactive phytochemicals of Centella asiatica under selected extraction conditions comparing the extraction behaviour of different solvent systems.
Is the liquid chromatography method based on any paper? Please reference it.
There were similar methods for the HPLC analyses of triperpenes in C. asiatica with some variation in the column dimensions and HPLC system used. The main reference [27] on which the current method was based, is reported under section 2.7 with the modifications used.
Check all the C. asiatica, they should be in italics, some of them are not
Checked. All are in italic now.
This kind of sentences ‘HPLC analyses of C. asiatica extracts are reported in the literature [26-32].’ Should be in the results and discuss section. Materials and methods section is only to describe the methods use. Please rephrase the sentence and indicate what methods you base your method. All the six methods (reference from 26-32 are necessary to reference?).
Point taken and the main reference on which the current method is based [27], was moved to the Methods Section and the other references were taken out. The sentence was rephrased.
Results
Could you please include a picture in the Table ST1 to complement the results?
A few photos were included as suggested by the Reviewer.
The figure S1 and S2 has not been attached to the submission or it is not available. Please include it.
They were sent with the manuscript when it was submitted first time. We will resend them.
This manuscript is a resubmission of an earlier submission. The following is a list of the peer review reports and author responses from that submission.
Round 1
Reviewer 1 Report
Comments and Suggestions for Authors
Dear authors
My suggestions and remarks are insterted in the attached file
Please revise carefully each point
Regards

Comments on the Quality of English Language
Reviewer 2 Report
Comments and Suggestions for Authors
The article titled ‘Total triterpenes, polyphenols, flavonoids, and antioxidant activity of bioactive phytochemicals of Centella asiatica by different extraction techniques’ has been reviewed. The article shows the description of four different extraction techniques applied to Centella asiatica plant in order to obtain bioactive compounds.
The lines are missing for reviewing
Introduction
Please provide a reference about the C. asiatica chemical composition (first line of the introduction)
The introduction section must be improved. You are mixing your aim with the introduction. The extraction techniques should be explained before your aim. I suggest including more extraction techniques, i.e. pressurized liquid extraction, supercritical fluid extraction. Also, UAW and MAE are not new anymore, they are considered conventional.
If your aim is to carry out the ‘investigation of different extraction methods for the efficient extraction of potential neuroprotective bioactive compounds’, where is the neutroprotective bioactive compounds description? What compounds are you looking for?
This sentence in introduction section ‘Analyses in the current study were performed by adopting some of these reported methods with a few modifications.’ should belong to materials and methods.
The aim is divided through the introduction section. First explain the state of the art and at the end, your objective.
Materials and methods
Explain the DNA molecular analysis method and the morphological characteristics method. Based on what method? Please include.
What particle size was after the ground step? Depending on the size the extraction yield and compounds extraction may vary.
‘1cm’ a space is missing
Is the liquid chromatography method based on any paper? Please reference it
Were the extractions carried out by duplicate, triplicate? Please add.
Results
The results are not well-explained, some information is mixed and tables are missing. The results and the discussion must be improved.
No tables were provided in the supplementary material
The figure 2 explanation is in the 3.1 or 3.2? Do not mix results
Is the figure 2 correct? There is no triterpenes detected?
The sentence ‘The variance was analyzed by one-way ANOVA followed by Tukey’s test to compare means, using GraphPad Prism version 8.3.1 for macOS (GraphPad Software, La Jolla California USA, www.graphpad.com).’ belongs to material and methods.
Why there is no information about SWE in Table 2
Figure S2 does not show that the compounds are extracted at the similar levels, it is only a chromatogram. The Table 2 might show they are extracted at the same level.
When you compare with other studies can you provide please some data? Amount of triterpenes content with same techniques or yield extraction percentage.
The percentage in the figure 3, and 4 (only the green graphic) are with two decimals. Please unify with the rest of the manuscript tables
Comments on the Quality of English Language
-